# Soil Contamination Assessments from Drilling Fluids and Produced Water Using Combined Field and Laboratory Investigations: A Case Study of Arkansas, USA

**DOI:** 10.3390/ijerph18052421

**Published:** 2021-03-02

**Authors:** Joshua Swigart, Joonghyeok Heo, Duane Wolf

**Affiliations:** 1Department of Geosciences, University of Texas Permian Basin, Odessa, TX 79762, USA; swigart_i@utpb.edu; 2Department of Crop, Soil, and Environmental Sciences, University of Arkansas, Fayetteville, AR 72701, USA; dwolf@ua.edu

**Keywords:** soil contamination, produced water, drilling fluid, Arkansas, field and laboratory investigations

## Abstract

Rotary drilling for oil and natural gas uses drilling fluid for lubrication of the bit, to seal off unstable shale layers, and floating out rock cuttings. Drilling fluid is a water–clay chemical mixture. Produced water is a water–sand chemical mixture. Land farming is a common disposal technique of drilling fluid and produced water. In the land farming process, amendments of fluid are repeatedly applied to the soil surface. Plant growth and soil chemical properties may be altered by additions of drilling fluid, because of alkalinity, salinity, trace elements, and petroleum residue contained in waste. The objective of this study was to determine the change in soil pH, electrical conductivity (EC), total nitrogen and carbon, and extractable nutrient levels following the land application of drilling fluid and produced water. The study was a comparison of three plots with similar soil properties and conditions. The three study plots had various levels of drilling fluid and produced water applications. The data show a major difference from field-to-field for EC, Na, and Cl levels. The EC and salt levels increased with additional applications of drilling fluid and produced water. The percent total nitrogen values and plant available P levels were very low in all fields. High EC and salt values, coupled with low N and P levels, would be detrimental to plant growth and development. To successfully vegetate this land-farm site, application of N and P fertilizer would be required. This study help to give a better understanding of practical ways to land-farm drilling fluid and produced water in a fashion that both minimizes environmental issues and is economically feasible in Arkansas. Thus, this research will provide important information for soil contamination management and contributes on understanding of the responses of soil properties to drilling fluid and produced water in the future.

## 1. Introduction

Increases in global petroleum and natural gas demand, coupled with new production technologies, have triggered increased rotary drilling to meet the world’s rising energy needs. Petroleum and gas wells are drilled in areas where it was not previously economically feasible. The United States is now the world’s largest oil producer. Rotary drilling is the common method of drilling implemented to reach petroleum and natural gas deposits [1,2]. In the rotary drilling process, a hole is drilled into the earth with a drilling rig that rotates a drill string with a bit attached. Drilling mud is used to lubricate the bit while drilling occurs [1,2]. The mud is pumped from the mud pits down the drill string to the bit. Nozzles then spray the bit, which cools and lubricates it. Drilling mud is also used to seal off porous geological strata and to stabilize shale layers. After the hole is drilled, sections of pipe are placed down-hole; this is called the casing. Cement is then, often, poured between the outside of the casing and the borehole. The casing ensures the structural integrity of the newly drilled well bore.

The drilling fluid is generally a water-based mud containing mainly bentonite clay. Most drilling muds consist of aqueous slurry of 5% bentonite in amended with sodium hydroxide and a density-increased material such as barite, to help float out rock cuttings [3]. The mud floats out the rock cuttings that are forced upward in the space between the drill pipe and down-hole wall. At the surface, the sediment and cuttings are separated. The mud is then sent to the mud tanks where additives, such as cotton seed hulls, are added to meet the appropriate specifications for the drilling project. Organic additives include petroleum products and compounds altered or man-made [4]. Inorganic additives consist of alkaline earth and metal salts employed to alter properties of bentonite clay [5,6,7]. Soil properties and plant growth can be negatively affected from by the contents of drilling fluid wastes. Drilling fluids often contain large amounts of salts that generally accumulate in soils. Miller and Pearson concluded that high levels of soluble salts or a high exchangeable sodium percentage (ESP) was a major cause of reduced plant growth. Chloride is essential for photosynthesis and serves other critical roles in plant functions [8,9,10]. Plants take up chloride as Cl^−^ anion, which is very mobile in the soil and is subject to leaching. In high concentrations, however, chlorides will inhibit plant growth, and they are specifically toxic to some plants. Excessive levels of chlorides in the soil can result in chloride-sensitive crops accumulating excessive amounts. The major detrimental effect of chloride to plants is its contribution to osmotic stress caused by excessive salts in the root zone [8,11]. Seedlings are usually more sensitive to soluble salts in the soil than are established plants. Soluble salts may accumulate in the soil causing saline conditions.

The volume of mud required to drill a hole is approximately three times the volume of the hole [12]. After the drill touches down at total depth, the primary drilling process is complete. After the drilling rig has moved to another site, the drilling fluid must be disposed of properly. Land application of waste, or land farming, has potential benefits such as reduction of waste mass and toxicity. In the land farming process, repeated applications to the soils surface occurs [13]. Land farming involves incorporation of drilling fluids into the soil surface [14,15]. The soil can also be tilled for aeration, and to help volatilize organic compounds; fertilizers are often added before incorporation. The process is designed to promote microbial degradation of the organic compounds in the drilling fluid and produced water. There are three main costs in land farming; these include labor to periodically till the soil, fertilizer additions, and maintenance of equipment. The equipment used in land farming, such as a tractor and plow, are commonly used in normal agricultural operations; monitoring for contaminants can be an additional cost. Nearby streams are often sampled and groundwater monitoring wells are used for sampling to determine in any contamination has occurred from the land farming operation.

The State of Arkansas has restrictions on contaminant levels being applied to landfarm soil surfaces, not the build-up of soil contamination levels in the soil profile. The Arkansas Department of Environmental Quality (ADEQ) is the decision-making body for State of Arkansas’s environmental regulations. Implementation of such regulations can include restrictions, such as not allowing drilling fluid to soils that have any pooled water already at the soil’s surface and not allowing drilling fluid applications that would cause ponding [16,17]. When drilling fluid is applied, the activity should be closely monitored to ensure the human error of over-applying and applying in unapproved areas does not occur. The ADEQ requires a 75% vegetation cover to bring a land-farm site back into compliance for site closure [17]. Sand, silt, and clay are the three components that determine soil texture. Multiple soil profile descriptions at a site can provide a great deal of information that may be useful in evaluating the variability of soil properties, and the directions and potential for transport of soil properties, and the directions and potential for contaminants in the subsurface [18,19]. Bentonite, along with other particles in the spent fluid can form surface crusts. Crusts can reduce infiltration capacity and hydraulic conductivity of the mud-amended soil [20]. The objective of this study was to determine the alteration of soil properties following application of drilling fluid and produced water: pH, Electrical Conductivity (EC), total nitrogen and carbon, and extractable nutrient content. The result contributes to understanding the impact of drilling fluid and produced water on soil contaminations. Therefore, this research contributes to an understanding of soil contamination and provides important information for oil and gas developments areas in making decision and development plans.

## 2. Study Area

### 2.1. Land-Farm Profile

The land farm in this study was located in Franklin County, 53 km east of Ft. Smith, AR. The coordinates of the site were 35°20′19.15″ N, 93° 56′46.34″ W. The study area, with a total area of 7.2 hectares, is in the western part of Arkansas (Table 1). The Arkansas River flows across the county from west to east. The total population was increased from 10,213 in 1960 to 18,125 in 2010 [21]. The county lays claim to the first oil strike in Arkansas and sits on vast fields of coal, clay, iron, shale, and other minerals; however, agriculture is its main economy base. The field study was a comparison of three fields at the study site: Fields 1, 2, and 3, which had similar soil properties and conditions (Figure 1). The three fields have varied levels of drilling fluid amendments, ranging from low-level (Field 1), a medium-level (Field 2), and a high amendment level (Field 3). The site manager reported that Field 3 received a higher rate of drilling fluid amendment than Field 2. Field 1 was approximately 2.4 ha and had the largest amount of vegetation. Field 2 was approximately 2 ha and had a large amount of vegetative cover; although there was less ground cover than that of Field 1. It was indicated that Field 1 had only 1 amendment of drilling fluid, while Fields 2 and 3 were amended multiple times. Field 3 was approximately 2.8 ha and received the highest amendment levels; it was largely non-vegetated, except for a small raised portion on the west side of the field. Vegetation at a site serves as an indicator of site history and site productivity and is a major determinant in erosion potential at a site [18]. Accurate records of amendment levels at the site were not kept. There were originally two settling ponds at the location; they have both been filled in and graded before the commencement of this project. There was also a stream that flowed approximately 30 m downhill south of Field 3 past the site, Hurricane Creek is part of the Arkansas River water shed [21].

### 2.2. Soil Characteristics

The Natural Resources Conservation Service (NRCS) Soil Survey indicated Fields 1 and 2 was a Leadvale silt loam that consists of deep to very deep, moderately well drained soils with a fragipan [22]. The soil consists of ~11, ~62, and ~27% sand, silt, and clay, respectively (Table 2); as determined by the hydrometer method. The hydrometer method is one of the most common methods for determining soil texture [21,22]. In the method, the percentage of sand, silt and clay is measured by using the United States Department of Agriculture (USDA) textual triangle. It is a fairly accurate method for determining the particle size distribution of a soil sample. It is fine-silty, siliceous, semi-active, and has Thermic Typic fragiudults. This soil is formed from material in uplands or local silty alluvium from nearby uplands underlain largely by shale, siltstone, sandstone, phyllite, and slate. Leadvale soil is located on slightly concave toe-slopes, benches, and terraces. Slope is an important site feature that influences the distribution of precipitation between the soil and surface run-off, and the movement of soil water [18,23]. The slope for a Leadvale soil is primarily less than 7% but can range from 0 to 15 percent. Fields 1 and 2 slopes from north-to-south from 135.6 m to 131.9 m and 138.6 m to 133.3 m feet above sea level, respectively. Field 3 is a Linker silt loam that consists of moderately deep, well drained, and moderately permeable soils. This soil is formed in loamy residuum weathered from sandstone. The soils are on broad plateaus, mountains, hilltops, and benches. Slopes are primarily 1 to 15 percent, but range to 30%. Field 3 slopes from northwest to southeast from 135 m to 130.1 m above sea level. The soil taxonomic class is fine-loamy, siliceous, semi-active, and thermic Typic Hapludents [22]. The soil had a sand, silt, and clay content of ~8, ~66, and ~26%, respectively.

## 3. Materials and Methods

A land farm used for disposal of freshwater and diesel contaminated drilling fluids and produced water is the focus for this study. Due to the fact that the study area is relatively small with an area of 7.2 hectare (0.072 km^2^), it is inevitable to use all the same fluid in the study area. For this reason, we believe that the same fluids were used in each field. The fluids generally consists of water, sand, guar gum, petroleum distillate, and hydrochloric acid [19,20]. From a scientific viewpoint, the purpose of sampling is to draw a collection of sampling units from a population mean without measuring all sampling units in the population [24]. Native soils are continuously variable and complex mixtures of gas, liquids, solids, and biota. After pre-approval by the site manager, sampling began at the three adjacent fields. Stratified random samples are obtained in a similar fashion as the simple random sample procedure except that the area to be sampled is broken into smaller subareas. Then each subarea is sampled following the simple random sample procedure previously described [24]. Each field was divided into subunits of approximately equal size (Figure 1). Within each sub-unit samples used a grid pattern with six locations; this was performed using a soil probe. Five, 2.54 cm-diameter cores were taken at a depth 0 to 15 cm. These samples were combined and mixed to form a composite sample for each sub-unit. The total of 36 samples were taken for three fields, control areas, and settling pond to support the field variability. The samples were also collected from the, drained, drilling fluid and produced water settling pond that was located on site (Figure 2).

After soil sampling was complete, samples were placed on ice in a cooler for transport back to the University of Arkansas at Fayetteville. Upon returning to the university, soil samples were air dried and ground using a mortar and pestle to pass a 2 mm sieve. Thirty-six samples were totally analyzed in the study area with a total area of 7.2 hectare (0.072 km^2^). Duplicate samples at locations 1A and 3D in Figure 1 were taken to measure range and Relative Percent Difference (RPD). The RPD is used as a quantitative indicator of quality assurance and quality control for repeated measurements where the outcome is expected to be the same and is calculated as a percentage. The procedures performed were pH, EC (1:2 soil ratio), total N and C by combustion with Elemental Veriomax, Environmental Protection Agency (EPA) Digestion Method 3050B, and (Mehlich-3 extractable) plant available nutrients (1:10) were P, K, Ca, Mg, S, Na, Fe Mn, Zn, Cu, and B [25]. Distilled water was used to extract Cl.

EPA Digestion Method 3050B was used to determine total P, K, Ca, Mg, S, Na, Fe, Mn, Zn, Cu, B, Al, Pb, Cr, Ni, and Ba. It is EPA’s selected analytical method for soil sample that is most widely applied in the US. The procedure comprises multiple cycles of high temperature digestion, evaporation, and cooling. Each digestion cycle the leachates were filtrated through 20 μm pore filter papers, diluted to 25 mL with ultrapure water, stored in 50 mL polypropylene centrifuge tubes, and refrigerated until compositional analysis. All samples were analyzed using Inductive Coupled Plasma–Atomic Emission Spectrometer (ICP-AES) at Soil Microbiology Laboratory and Agricultural Diagnostic Laboratory in the University of Arkansas, Fayetteville, USA (Figure 3). Upon completion of soil analysis, the data were analyzed to find the average soil nutrient content for each field and the onsite ponded drilling mud. The limit of detection (LOD) is the lowest concentration that can be detected by a method. The limit of quantitation (LOQ) is the lowest concentration at which the sample can be quantitated at defined levels for accuracy. In the method, the LOD has been defined as the lowest concentration tested that is equal to the average of a blank sample. The LOQ is defined as the lowest concentration where the coefficient of variation is less than 10%. The linearity of the method is the ability to generate results proportional to the concentration in the sample. The sensibility (*S*) shows the variation of the read response versus concentration of sample. Prior to generating a final reports, the data is reviewed a second time. The second review includes a sensibility check and technical criteria.

## 4. Results and Discussion

There were two previous greenhouse studies associated with this landfarm site. An undergraduate research study was performed looking at “Using Soil Amendments to Increase Bermuda Grass Growth in Soil Contaminated with Hydraulic Fracturing Drilling Fluid” [26]. This was a 9-week, 12-h of daylight, greenhouse experiment conducted between 19 January 2012, and 30 March 2012. In this study, varied levels of contaminated soil was collected in December 2011 from the 0 to 15 cm and 0 to 30 cm in Field 3. Organic amendments of broiler litter and Milorganite were characterized for their initial physical and chemical properties. Appropriate amendments were added and thoroughly mixed according to recommendations from the University of Arkansas—Agriculture’s Cooperative Extension Service. Inorganic fertilizer amendments of ammonium nitrate (34-0-0) were used, as well as phosphorous, as triple super phosphate (0-46-0); potassium levels were at optimal.

Based on the results of this study, the addition of the recommended plant nutrients enhanced Bermuda grass growth. After the nine-week plant growth phase, soil EC and water-extractable Cl were greater in the 0–15 cm depth soil compared to the 0–30 cm depth for the respective vegetation treatments [26]. They found that in the Bermuda grass study of the 0–30 cm depth soil had lower EC and water-extractable Cl levels than the non-vegetated treatment because of Bermuda grass plant uptake. In addition, the mixing of the surface-applied produced water with the 0–30 cm soil depth resulted in a dilution effect that decreased detrimental soil salinity effects. Bermuda grass shoot Na and Cl concentrations were unaffected by soil depth interval or the addition of soil amendments. The addition of plant nutrients from synthetic or organic soil amendments resulted in greater shoot biomass. Milorganite-amended soil had a greater extractable Na concentration than the inorganic fertilizer treatment for soil from the 0 to 30 cm depth. A study was performed to determine the “Effect of Drilling Mud on Plant Growth, Plant Chemical Properties, and Soil Chemical Properties” [27]. In the six-week greenhouse study, two plant species were grown in a Roxana loam soil amended with three rates of drilling-mud amended soil. Soil was collected from the remaining drained settling pond on site; large columnar structures were visible, with some cracks as visibly deep as six feet.

### 4.1. Drilling Mud Results

Soil samples taken from a drained settling pond located on-site revealed many of the same characteristics as soil samples taken from fields on-site [7,20]. The ADEQ limits drilling fluid application of pH levels to between 6.0 and 9.0: average values in this study ranged from 8.0 in Field 3 to 8.3 in Field 1, which is an allowable level. The pH from actual drilling mud from the drained settling pond on-site are nearly the same as that of the soils on site (Table 3). This is a clear indicator of drilling fluid affecting soil pH. EC levels in settling pond mud was very high at 6340 mg/kg, which indicates high salt levels in drilling fluid which have settled out and built up in the mud at the bottom of the pond. Na and Cl also tested very high at 5017 mg/kg and 6410 mg/kg, respectively. Ca levels also tested over allowable limits for application to soils. Excessive salts in soils are detrimental to optimum plant growth and can slow or inhibit plant growth in general. Salts accumulate in the root zone which negatively affects plant growth. Accumulation of excess salts in the root zone can hinder a plants ability to withdraw water. Regardless of available water, levels that can be taken up by the plant decrease. Available salts in water cause plants to exert more energy to up-take water, causing plant stress.

### 4.2. Soil Analysis

Evidence of hydrocarbons was most evident in the areas furthest down-gradient. Field 3 soils were blackened, and hydrocarbon odor was noticed [28,29]. EC levels increased substantially with additional applications of drilling fluid, where Field 1 had the lowest values and Field 3 had the highest (Table 4 and Figure 4). Field 3 soil EC values were more than twice the limit allowed for drilling fluid. There was a peak in Field 3 EC at 2.6 dS/m, as was expected, because the data from the drilling fluid settling pond mud had an EC of 6.42 dS/m (Table 4) and the control area of 0.06 ds/m. These EC values are more than the 1.0 dS/m allowed for a fluid at the time of application. EC is a measurement of the dissolved material in an aqueous solution, which refers to the ability of material to conduct an electrical current. EC is an important indicator of soil health because it indicates how much dissolved substances, chemicals, and minerals are present in the material. Higher amounts of these impurities will lead to a higher conductivity. Thus, EC is a direct indicator of soil salinity [11]. As was expected, Na and Cl results inhibited a pattern similar to that of EC, where Field 3 values were higher than 1 or 2 (Figure 5). Arkansas law allows for land application of drilling fluids if soil Cl levels are below 1000 mg/kg in land farmed soils; Field 3′s mean value Cl is 2165 mg/kg. The heavy metals Ni, Cu, and Zn mean values in this study were all substantially below levels allowed by law (Figure 6). For Fields 1, 2, and 3, the pH data showed little variability with additional drilling fluid applications (Table 3).

The percent total nitrogen values were low in Fields 1, 2, and 3, which is too low to promote optimum plant growth (Figure 7). Field soil test P levels were at or below 4.1 mg/kg in all drilling fluid amended fields, which is also detrimental to plant growth and vitality. Inorganic forms of phosphorous occur in combination with iron, aluminum, calcium, fluorine, or other elements [30]. The K levels were uniform in all fields and was sufficient for plant growth. Field average Mg, S, Cu, and Zn levels were suitable to meet plant needs in all fields (Table 3). Field nutrient levels should be brought up to optimum for all fields in this study. Growth was noticeably better where fertilizer had been applied and where fertilizer was applied in combination with surface mulch or manure (not incorporated into the soil) [31,32]. Analysis for percent C in soils is an important aspect of contaminant delineation. Total carbon percentages seam to increase with applications of drilling fluid and produced water in this study (Figure 8). Fields 1, 2, and 3 had percent total carbon values of 1.23, 1.33, and 2.73, respectively; and the control area had 0.73% C. Higher percent total carbon content is a direct indicator of increased levels of hydrocarbons in the soil, possibly the result of contamination.

EPA digestion Method 3050B displaced Pb and As levels as undetectable in all field’s samples (Table 5). Field average Al and Fe totals were elevated in all fields (Table 5). The total digestion testing showed P, K, Mg, S, NA, Mn, and B levels were highest in Field 3, as was expected. Na concentrations were elevated in Field 3 and the mud from the settling pond at 1726 mg/kg and 5017 mg/kg, respectively. Sodium dispersion causes infiltration and hydraulic conductivity to be reduced and crusting at the soil surface. Natural binding of clay particles is impeded when sodium ions block them from binding. Swelling and soil dispersion is then caused by clay particles expanding. Permeability reduces when clay particles plug soil pores as a result of soil dispersion. A hard crust can then result from clay dispersion after repeated wetting and drying.

All RPD values tested, except percent total N, fell under fifteen percent, which means that section composite samples were nearly uniform (Table 6). In Section 1A in Figure 1, RPD values were uniform for pH and B, at zero percent (Table 6). Soil aluminum levels were elevated in all fields which can be detrimental to plant growth and vitality. Excessive aluminum levels in soil can cause damage to plant roots. When damaged root systems occur, symptoms above ground are likely. Aluminum and phosphorous compounds can develop in soil, causing a phosphorous deficiency (Figure 9). Absorption of water can be reduced by poor root development. Aluminum–sulfur compounds can also develop, reducing availability of sulfur. Reduction of availability of other nutrient cations can also occur through competitive interaction. Aluminum is not an essential element for plant growth, although it makes up seven percent of the mass of the earth’s crust. Barium levels were approximately in all fields were 20-fold greater, compared to the control area. Iron levels appear to be in the normal range, with a site average of 22,107 mg/kg (Figure 10). The typical range of iron concentrations in soils is from 0.2% to 55% (20,000 to 550,000 mg/kg) according to Bodek et al. [33]. Large amounts of iron can be released during the coal mining process. Mean concentrations of Zn, Cu, B, and Ni were similar in all fields.

### 4.3. Field Variability

Soil analysis values varied largely within each field indicating that there was a large amount of variability between applications of drilling fluid and produced water. Mehlich 3 extractable sodium and chloride values in Field 3 ranged from 1344 to 2072 mg/kg and 1434 to 3026 mg/kg, respectively. EPA digestion Method 3050B values in Field 1 for iron and aluminum ranged from 20,895 to 26,618 mg/kg and 9140 to 12,960 mg/kg, respectively (Figure 10). Accurate records were not kept, and it hypothesized that there was unlikely an attempt to spread fluids uniformly across each field on site or from field to field. It is further hypothesized that Field 3′s mean elevated levels of contaminants occurred as a result of the field being longer than all other fields which allowed the truck drivers applying fluid to the soil surface to be able to apply fluids in one “pass”; doing this reduced their total turn-around time by allowing them to reduce their unloading time. Soil textural analysis varied from field-to-field and controls’ east and west [34].

Composite sampling was performed during a return trip to the site in January 2020 for Fields 1, 2, 3, and Control Samples East and West to look for textural variability between fields. The analysis for texture of samples from the site showed varied amounts of sand, silt, and clay between Fields 1, 2, and 3, and the two control samples (Table 1). Fields 1, 2, and 3 had higher values of sand than that of Control Samples East and West at 11.5%, 10.4%, 7.8%, 5.6%, and 5.6%, respectively (Table 1). A higher percentage of sand was expected in the fields applied, than that of control samples, as vast amounts of sand are used during the fracturing process to hold open fractures in underlying rock. It was not expected that Field 1 would have the highest levels of sand, followed by Fields 2 then 3. It is hypothesized that because the settling ponds up-gradient were drained and graded, large amounts of sand were applied to the soil surface of the fields nearest the settling ponds (Fields 1 and 2). Clay percentages were highest in Field 1, followed by 2 and 3 at 27.8%, 26.5%, 26.5%, 17.6, and 16.4, respectively. Clay percentages were much higher in Fields 1, 2, and 3, than that of Controls East and West and is mostly likely a result of mud additions during the drilling process. Silt percentages of Fields 1, 2, 3, and Control East and West showed results of silt to be lowest in Field 1 and highest in Control West at 60.7%, 63.1%, 65.7%, 76.8%, and 78.0%. Higher silt percentages were expected in control samples, as silt additions do not occur during the drilling and fracturing processes.

### 4.4. Contamination with Fluids

The exploration of oil and natural gas requires the use of drilling fluids. Drilling fluids are the materials created for the purpose of drilling oil and natural gas wells. The fluids are pumped into the hole during the drilling process to help cool and lubricate the bit, suspend cuttings, seal the formation, and control well bore pressure. Drilling mud is continuously recycled to remove solids until it can no longer be utilized. After drilling is completed, the drilling fluid and mud in the reserve pit must be disposed of. Land application or land-farming is a generally accepted method of disposing of the contents of the reserve pit. It is a waste management practice in which oil and gas wastes are mixed with or applied to the land surface. Due to the increase in drilling in our study area, the need of drilling fluids through land application has increased (Figure 11). Consequently, there are potential contamination associated with land application of the fluids.

Contamination of soil or water sources off-site appear to be highly unlikely. There was a 30-m wide buffer zone between Field 3 and Hurricane Creek, which consisted of native tall grasses, which should eliminate the possibility of contaminants moving down gradient and into the waterway. Baseline soil samples were not taken prior to the beginning of the land-farm process; this makes it difficult to speculate whether there was possible contamination that already existed on site as a result of natural processes. Nickel concentrations were normal in all fields with a mean of 14.97 mg/kg; soil ranges from 10 to 1000 mg/kg are normal.

## 5. Conclusions

Drilling fluid applications to soil results in changes in soil chemical properties. These effects were most visibly apparent in Field 3; which was largely un-vegetated at the original time of sampling. The EC levels were highest in Field 3, where Na and Cl had accumulated in the surface soil over time; the effects were detrimental to the soil. Precipitation was needed to push salts leach into the soil profile. The total N and extractable P levels, which are essential nutrients for plant growth, were low in all fields. The previous sprigging of Bermuda grass, in 2009, by company employees was the appropriate method of vegetation establishment, but was not effective, probably because of lack of precipitation after sprigging and the timing of the event was late summer and field N and P levels were inadequate. The fields in the study needed precipitation, supplemented by irrigation, to promote plant growth; these fields were re-sprigged in Spring 2010.

Site closure procedures requires vegetative coverage of 75% or more, or equivalent to the surrounding landscape, whichever is less, within six months of site closure. Arkansas does not have standard guidelines for allowable soil contaminant levels of land-farms. Instead the state uses standards set in Regulation 23 for the management of remediation and related wastes, usually arriving at a site-specific standard for each clean-up. There were elevated levels of Na, Cl, and Ba within each field; this is likely the result of over application, in addition to spreading fluids that exceeded allowable contaminant limits. After revegetation occurred, the land-farm was then decommissioned and used for agricultural purposes; hay farming and cattle grazing. Visual results of a return trip to the location, in January 2020, showed nearly complete vegetation coverage of Fields 1, 2, and 3. Previous remediation recommendations made were implemented to bring the location towards satisfactory vegetation levels and closure. A return trip occurred in January 2020 where nearly 100% vegetative cover was viewed. This research contributes on understanding of the responses of soil properties associated with energy developments in Arkansas. Therefore, this research can provide important information for soil contamination manager in making important decision and developing plans for use of the soil resources in the future.

## Figures and Tables

**Figure 1 ijerph-18-02421-f001:**
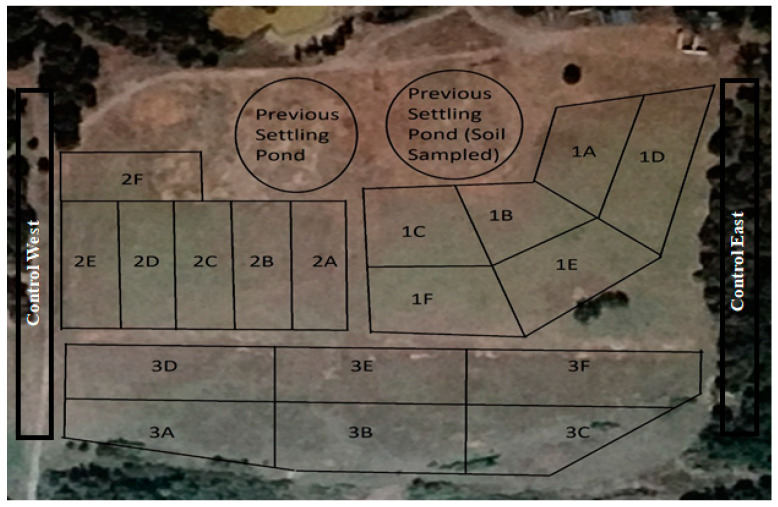
Map of the land-farm site showing three fields sampled, control areas, and settling pond. Letters indicate subunits sampled in each field.

**Figure 2 ijerph-18-02421-f002:**
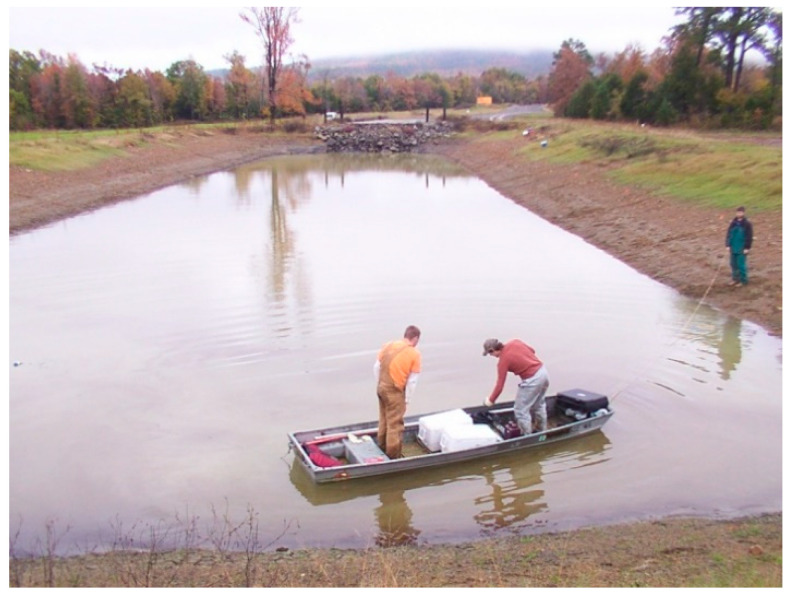
One settling pond used for storage in the study area.

**Figure 3 ijerph-18-02421-f003:**
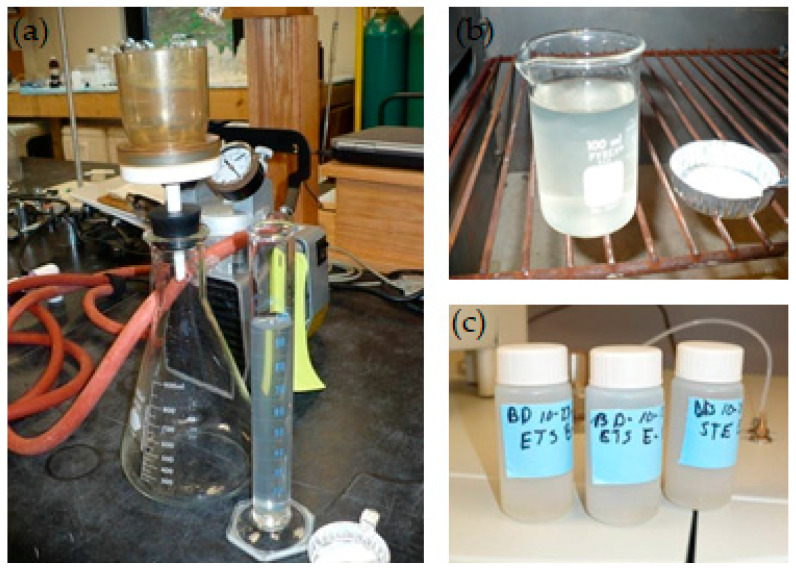
Samples taken in wooded area east of Field 1, 2, and 3 for the study area: (**a**) sample preparation, (**b**) digested solution and (**c**) sample bottle.

**Figure 4 ijerph-18-02421-f004:**
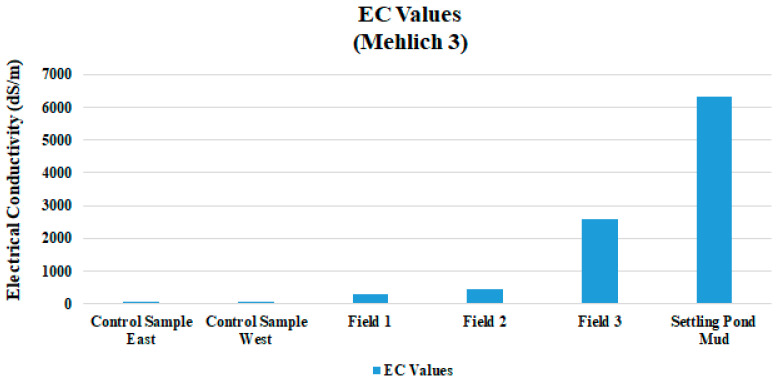
EC levels for three fields, two controls, and one settling pond sampled for study.

**Figure 5 ijerph-18-02421-f005:**
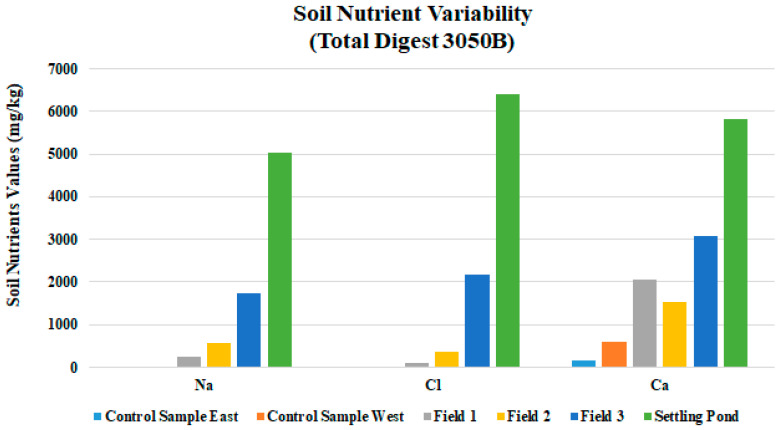
Mean sodium (Na), chloride (Cl), and calcium (Ca) values for three fields sampled during the study.

**Figure 6 ijerph-18-02421-f006:**
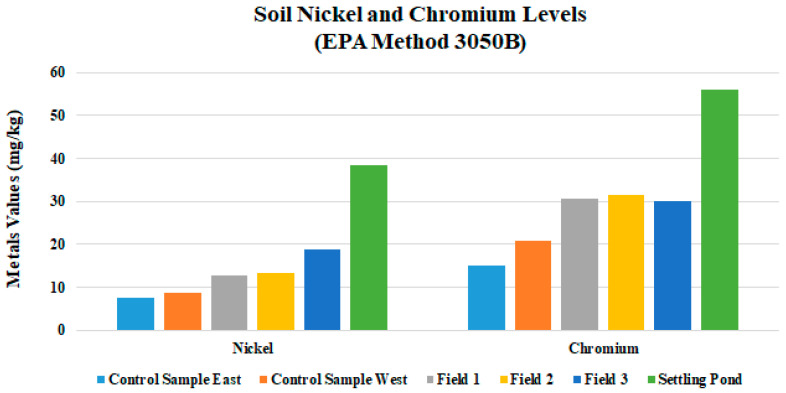
Mean values for nickel and chromium of Fields 1, 2, and 3 and Control Samples East and West. EPA: Environmental Protection Agency

**Figure 7 ijerph-18-02421-f007:**
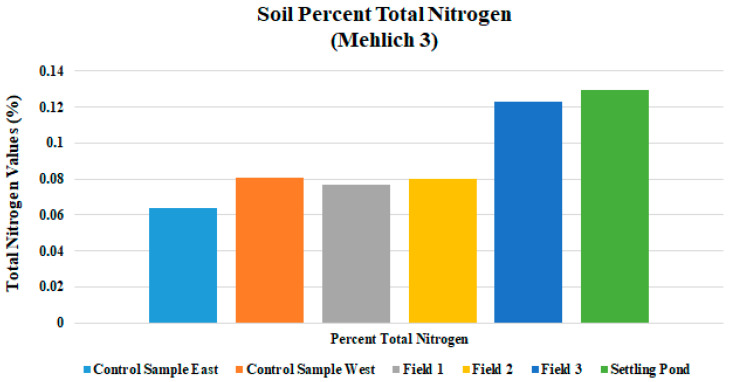
Mean total nitrogen levels for three fields and three sample ponds sampled during the study.

**Figure 8 ijerph-18-02421-f008:**
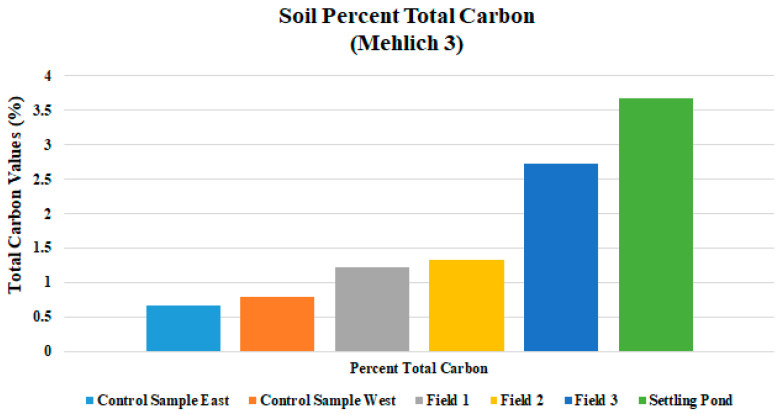
Mean total carbon levels for three fields and three sample ponds.

**Figure 9 ijerph-18-02421-f009:**
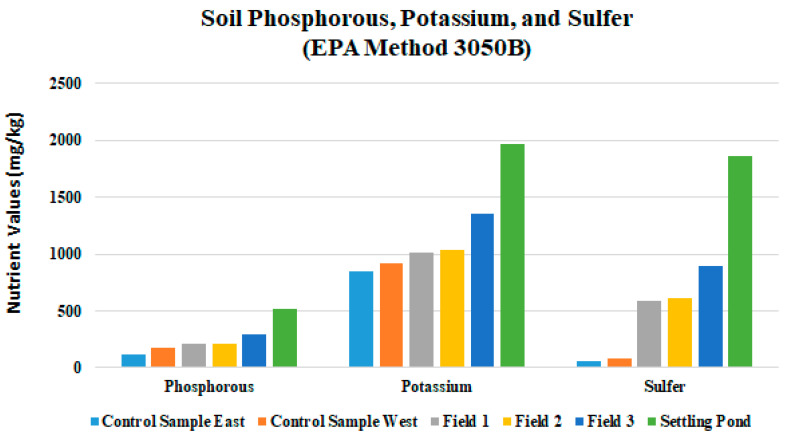
Soil phosphorous, potassium, and sulfur of Fields 1, 2, and 3 and Control Samples East and West.

**Figure 10 ijerph-18-02421-f010:**
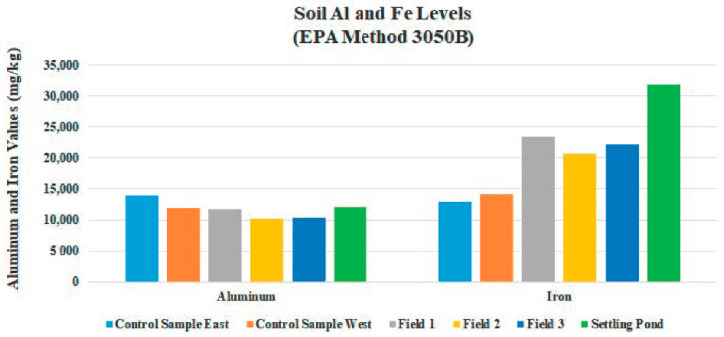
Mean values for aluminum and iron of Fields 1, 2, and 3 and Control Samples East and West.

**Figure 11 ijerph-18-02421-f011:**
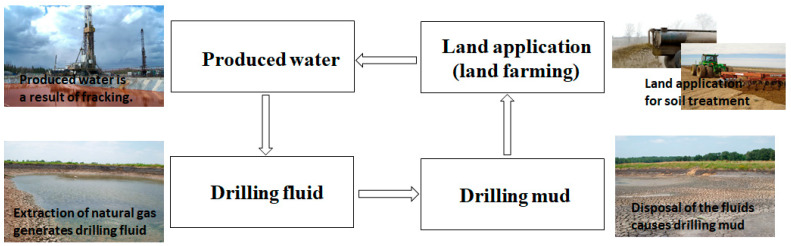
The diagram for drilling fluid and land affections cycling.

**Table 1 ijerph-18-02421-t001:** The summary of the land-farm description in the study area.

Land-Farm Description	Field 1	Field 2	Field 3
Location	Franklin County, AR (35°20′19.15″ N, 93° 56′46.34″ W)		
Surface area (hectare)	2.4	2.0	2.8
Levels of drilling fluids	Low-level	Medium-level	High-level
Vegetation cover	Largest	Intermediate	Non

**Table 2 ijerph-18-02421-t002:** Soil characteristics in the study area.

Soil Characteristics	Field 1	Field 2	Field 3
Sand (%)	11.5	10.4	7.8
Silt (%)	60.7	63.1	65.7
Clay (%)	27.8	26.5	26.5
Minerals formation	Shale, Siltstone, Sandstone, Phyllite, Slate		
Bulk density	0.59–1.17 Mg m^−3^		

**Table 3 ijerph-18-02421-t003:** The pH, total, and Mehlich-3 extractable elemental levels, % total N and C, and Cl levels of soil from a drained drilling fluid and produced water settling pond. Mean of two soil composite samples collected from a drilling fluid and produced water settling pond.

P	K	Ca	Mg	S	Na	Fe	Mn	Zn	Cu	B	Cl	Al	Pb	Cr	Ni	Ba	C	N	pH	EC
Mehlich-3 (mg/kg)																	%		s.u. dS/m	
3.3	289	5767	513	461	4980	649	99	10.4	11	4.7	6425	N/A	N/A	N/A	N/A	N/A	3.7	0.13	8.0	6420
Total Elemental (mg/kg)																				
521	1958	17,825	4263	1823	5335	32,255	762	93	59	13	N/A	12,125	111	57	38	1988	N/A	N/A	N/A	N/A

s.u.: Standard unit, N/A: not available.

**Table 4 ijerph-18-02421-t004:** The pH, electrical conductivity (EC), Mehlich-3 extractable nutrient levels (mg/kg), % total N and C, and chloride levels (mg/kg) of three fields at the study site. Mean of six composite soil samples collected in each field (SD: Standard deviation).

Parameters	Field 1		Field 2		Field 3	
	Mean	SD	Mean	SD	Mean	SD
pH Units	8.3	±0.3	8.2	±0.2	8	±0.3
1 Soil:2 Water Ratio						
EC dS/m	0.314	0.12	0.465	0.21	2.589	0.618
Mehlich-3 Extractable Nutrient Levels						
P	4.1	0.7	3.5	0.2	3.9	0.2
K	188	20	186	15	186	26
Ca	2069	568	1526	275	3068	400
Mg	163	15	182	29	309	30
S	38.8	34.5	29.5	10.2	78.1	11.2
Na	260	82	563	167	1726	293
Fe	204	37	252	52	409	40
Mn	104	16	130	29	140	39
Zn	4.5	0.8	5.4	1.3	8.8	1
Cu	3.7	0.5	3.8	0.8	6.3	1
B	1.6	0.2	2.1	0.3	3.1	0.6
Water Extractable						
Cl	98	45	380	234	2165	666
Total C and N Levels						
Total N %	0.077	0.007	0.08	0.008	0.123	0.012
Total C %	1.231	0.173	1.328	0.233	2.726	0.304

**Table 5 ijerph-18-02421-t005:** Total elemental levels (mg/kg) of three fields at the study site analyzed by EPA digestion Method 3050B. Mean of six composite soil samples collected in each field (SD: standard deviation).

Parameters	Field 1		Field 2		Field 3	
	Mean	SD	Mean	SD	Mean	SD
S	588	52	646	71	894	57
Na	351	97	608	165	1823	284
Fe	23,653	97	20,658	2007	22,274	1342
Mn	331	46	409	79	627	96
Zn	35.7	1.8	37.5	5.8	53.3	6
Cu	12.1	1	11.6	1.6	17.1	2.1
B	8.8	0.7	8.7	0.8	10.7	0.7
Al	11,881	1503	380	234	2165	666
Cr	30.9	3.9	31.5	7.7	30.2	2.8
Ni	12.9	0.3	13.3	1.4	18.8	1.9
Ba	2272	86	2416	174	2463	63

**Table 6 ijerph-18-02421-t006:** Fields 1, 2, and 3 soil sample and duplicate range and Relative Percent Difference (RPD). Soil Sample nutrient data from the research site.

Sample Description	Mehlich-3															
	pH	EC	P	K	Ca	Mg	S	Na	Fe	Mn	Zn	Cu	B	Cl	%Total N	% Total C
**Units**	**ds/m**	**mg/kg**													**%**	
1A and 1A Duplicate																
Range	0	0.34	−0.5	20	−24	−5	−4.5	−32	4	5	−0.1	−0.1	0	11	0.03	0.052
RPD	0	−11.6	−11.8	10.3	−1.6	−3.3	35.6	−8.4	2.3	5.1	2.5	2.7	0	12.3	4.1	4.4
3D and 3D Duplicate																
Range	0.1	0.36	0.2	6	316	0	1	−140	37	9	0.6	0.3	0.1	382	0.0175	0.526
RPD	1.3	12.1	5.1	1.9	10.6	0	1.35	−7	9.3	5.7	−7	5.3	3.6	13.5	14.8	20
	**EPA Method 3050B**															
	**P**	**K**	**Ca**	**Mg**	**S**	**Na**	**Fe**	**Mn**	**Zn**	**Cu**	**B**	**Al**	**Cr**	**Ni**	**Ba**	
1A and 1A Duplicate																
Range	−24.7	−15.55	−274	−56	−41	−48	−1202	−27	3	−0.6	−0.2	−771	−3.4	−0.3	117	
RPD	−10.7	−1.53	−12.8	−5.4	−7.5	−10.7	−4.7	−8.6	8.5	−4.9	−2	−6.21	−9.4	2.3	−5.4	
3D and 3D Duplicate																
Range	−12	−4	39	41	94	−48	952	9	4	−0.6	−0.2	−220	3.9	−0.2	68	
RPD	−4.38	−0.3	−0.8	2.3	10.8	−2.7	4.5	1.4	8	−3.8	−2	−2.26	14	−1.2	2.8	

## Data Availability

Data sharing not applicable (analyses were performed at the Soil Microbiology Laboratory and Agricultural Diagnostic Laboratory in University of Arkansas, Fayetteville, AR, USA).

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
