# Peer review of "Soil Contamination Assessments from Drilling Fluids and Produced Water Using Combined Field and Laboratory Investigations: A Case Study of Arkansas, USA"

_ijerph, 2021, doi:10.3390/ijerph18052421_

Round 1

Reviewer 1 Report

The study deals with an important issues, however after reading the title the reader would expect much more, at least something different. "The comparison of environmental effects from oil and gas produced fluids" is very misleading. The overall impression is that the manuscript is rather a case study than a scientific research, so the title must be changed. The major concern is about the unsufficient amount of initial data to provide valuable results. The effects described within manuscript can be a result of many factors. For example landform is not taken into account, microclimate and local atmospheric conditions are missing. The exact parameters of the fluids are also unknown. Are the authors convinced that the fluids were all the same at each field? For a case study these issues should be mentioned and discussed, if calling it scientific research, a model soil and fluid should be used for validation and confirmation of the results.

Reviewer 2 Report

Minor drawbacks and recommended improvements

1

Line 29, page 1

The Keywords Environmental Effects, oil, and gas are in the title. I suggest that the authors remove these keywords from the title or replace them with other keywords words.

2

Line 35, page 1

the authors should indicate in this paragraph the reference of this method.

3

Line 43, page 2

The authors should incorporate the word “mainly” in the phrase “The drilling fluid is generally a water-based mud containing mainly bentonite clay”. In this sense, I suggest removing the phrase “The main component of drilling fluid is bentonite clay” in line 85, page 2.

4

Line 92-102, page 3

The authors should indicate the land farm description in a table, such as location, field (1, 2, or 3), land farm surface, levels of drilling fluids amendment, amount of vegetation. This table may be cited in the results and discussion section to facilitate the discussion of results.

5

Line 111-125, page 4

The authors should indicate the soil's physical chemistry characteristics from this study, such as texture (sand, silt, and clay content), minerals, bulk density, taxonomy, etc. in a table. This table may be cited in the results and discussion section to facilitate the discussion of results.

6

Line 113, page 4

The authors should indicate the methods used.

7

Line 213, page 6

Line 218, page 7

The authors defined the abbreviature to EC in Line 89, page 3.

Major drawbacks and recommended improvements

8

Line 154, page 5

In relation to the analytical methodology to determine total P, K, Ca, Mg, S, Na, Fe, Mn, Zn, Cu, B, Al, Pb, Cr, Ni, and Ba, the authors should indicate the analytical technique used in this study and the quality parameters (limit of detection (LOD), limit of quantification (LOQ), sensibility (S) and linearity).

Reviewer 3 Report

Oil and natural gas drilling fluids contain mixed water,clay, sand and chemicales. Their previous disposal has bad effects on land farming. How to make an amendments of drilling fluids applied to the soil surface will have direct affect on envirormnet, plant growth, soil chemical properties  from their alkalinity, salinity, trace elements, and petroleum residue contained in waste. Thus the significance of the manuscript is high. The issues are proposed here for considerations.

  1. The paper title may be of too wide content - Comparison of Environmental Effects from Oil and Gas  Produced Fluids using Combined Field and Laboratory Investigations.  That is, narrow title is proper
  2. The said oil and gas drilling fluids in the said county may require some  information of basic backgrounds, data or componnents, etc.
  3. It would be better, the oil drilling fluid and gas fluid in different separate investigatiions. 
  4. the total samples of 6 in 1-3 fields seem not be enough. Some other samples required. 
  5. That is , the kind of drilling fluid - Factor level- land soil affections diagram may be added.   

Reviewer 4 Report

The paper entitled “Comparison of environmental effects from oil and gas produced fluids using combined field and laboratory investigations” presents how the soil chemical    properties change through the application of drilling fluid and the produced water. An extensive study is presented in all different investigated study fields. All the results are well established. However, the main concern of the reviewer is the novelty and the contribution of this research. How this research provides an added scientific value in the field and what is the main contribution? The reviewer thinks that the contribution of this work is marginal thus it should be reconsider for publication after major revision.

Round 2

Reviewer 2 Report

Manuscript Number: ijerph-1042339

Minor drawbacks and recommended improvements

1

Line 29, page 1

The Keywords Environmental Effects, oil, and gas are in the title. I suggest that the authors remove these keywords from the title or replace them with other keywords words.

The authors removed the Keywords “Environmental Effects, oil, and gas” from the keywords (refer to Line 30 page 1 in the edited version).

2

Line 35, page 1

the authors should indicate in this paragraph the reference of this method.

The authors incorporated this suggestion into the revised version (line 38, page 2).

3

Line 43, page 2

The authors should incorporate the word “mainly” in the phrase “The drilling fluid is generally a water-based mud containing mainly bentonite clay”. In this sense, I suggest removing the phrase “The main component of drilling fluid is bentonite clay” in line 85, page 2.

The authors incorporated this suggestion into the revised version (line 45, page 2).

4

Line 92-102, page 3

The authors should indicate the land farm description in a table, such as location, field (1, 2, or 3), land farm surface, levels of drilling fluids amendment, amount of vegetation. This table may be cited in the results and discussion section to facilitate the discussion of results.

The authors incorporated this suggestion into the revised version (line 98, page 3 and Line 120, page 4).

5

Line 111-125, page 4

The authors should indicate the soil's physical chemistry characteristics from this study, such as texture (sand, silt, and clay content), minerals, bulk density, taxonomy, etc. in a table. This table may be cited in the results and discussion section to facilitate the discussion of results.

The authors incorporated this suggestion into the revised version (line126, page 4 and Line 143, page 5).

6

Line 113, page 4

The authors should indicate the methods used.

7

Line 213, page 6

Line 218, page 7

The authors defined the abbreviature to EC in Line 89, page 3.

Major drawbacks and recommended improvements

8

Line 154, page 5

In relation to the analytical methodology to determine total P, K, Ca, Mg, S, Na, Fe, Mn, Zn, Cu, B, Al, Pb, Cr, Ni, and Ba, the authors should indicate the analytical technique used in this study and the quality parameters (limit of detection (LOD), limit of quantification (LOQ), sensibility (S) and linearity).

The authors incorporate the analytical technique used in this study but do not incorporate the quality parameters (limit of detection (LOD), the limit of quantification (LOQ), sensibility (S), and linearity).

Reviewer 4 Report

The manuscript has been improved as the reviewer suggested and the novelty and the contribution of the present study has been clearly defined. Therefore, the reviewer suggest this research work acceptable for publication in the present form.
